# CD163 as a Potential Biomarker in Colorectal Cancer for Tumor Microenvironment and Cancer Prognosis: A Swedish Study from Tissue Microarrays to Big Data Analyses

**DOI:** 10.3390/cancers14246166

**Published:** 2022-12-14

**Authors:** Shuwen Ma, Yuxin Zhao, Xingyi Liu, Alexander Sun Zhang, Hong Zhang, Guang Hu, Xiao-Feng Sun

**Affiliations:** 1Institute of Environmental Medicine, Karolinska Institute, SE-171 77 Stockholm, Sweden; 2Department of Epidemiology, School of Public Health, China Medical University, Shenyang 110122, China; 3Centre for Systems Biology, Department of Bioinformatics, School of Biology and Basic Medical Sciences, Soochow University, Suzhou 215006, China; 4Department of Oncology-Pathology, Karolinska Institute, SE-171 77 Stockholm, Sweden; 5School of Medical Sciences, Faculty of Medicine and Health, Örebro University, SE-701 82 Örebro, Sweden; 6Department of Oncology, and Department of Biomedical and Clinical Sciences, Linköping University, SE-581 83 Linköping, Sweden

**Keywords:** CD163, TME, TCGA, prognosis, CRC

## Abstract

**Simple Summary:**

Through the analysis of tissue microarray (TMA) samples from colorectal cancer (CRC) patients and bioinformatical analyses of public databases and our clinical dataset, this study identifies the different expressions of CD163 in various tissues, the presence of the receptor in TME, the interaction with other biological processes and a positive correlation between CD163 dysfunction and worse prognosis. Therefore, CD163 can be used as a new biomarker to predict patient prognosis.

**Abstract:**

(1) Background: CD163, a specific macrophage receptor, affects the progression of malignant tumors. Unfortunately, the regulation and expression of CD163 are poorly understood. In this study, we determined the expressions of CD163 in TMA samples from CRC patients and combined them with patient data from several Swedish hospitals. (2) Methods: The expressions of CD163 in tissue samples from CRC patients were examined. After combining 472 CRC patients’ gene expression and 438 CRC patients’ clinical data with the TCGA database, 964 cases from the GEO database, and experimental expression data from 1247 Swedish CRC patients, we selected four genes (PCNA, LOX, BCL2, and CD163) and analyzed the tumor-infiltrating immune cells (TICs) and CRC prognosis. (3) Results: Based on histopathological TMA analysis, CD163 was strongly expressed in the stroma of both normal and cancer tissues, and the expressions in normal and cancer cells varied from negative to strong. The results from public databases show decreased expression of CD163 in cancer tissue compared to normal mucosa (|log FC| > 1 and FDR < 0.01), and it is a negative prognostic factor for CRC patients (*p*-value < 0.05). Through tumor microenvironment (TME) analysis, we found a potential influence of CD163 on immune cell infiltration. Furthermore, the enrichment analysis indicated the possible interaction with other proteins and biological pathways. (4) Conclusions: CD163 is expressed differently in CRC tissue and is a negative prognostic factor. Its expression is associated with the TME and tumor purity of CRC. Considering all results, CD163 has the potential to be a predictive biomarker in the investigation of CRC.

## 1. Introduction 

Colorectal cancer (CRC) is the second most common malignant tumor in women and the third most common in men. Approximately 900,000 deaths are attributed to this cancer each year, making it the cancer with the fourth highest mortality. Therefore, CRC is an important public health problem [1,2]. According to the 2020 statistical cancer report from the American Cancer Society [3] there have been improvements in early diagnosis, treatment response, and survival rate of CRC patients in recent years. However, many patients are still diagnosed at late stages of the disease, and around 30% of cancers are still resistant to common therapy, leading to poor prognosis. Although there are many new anticancer drugs in development, such as glutamic acid sulfonamides [4], there are still not enough powerful anticancer drugs in clinical trials [5]. One of the main reasons for clinical outcomes and poor prognosis is that the currently used screening programs and clinicopathological tools cannot detect a tumor at its early stages nor be used to develop personalized treatment regimes. Thus, it is urgently required to find promising biomarkers for approaching diagnosis and precision medicine.

Research has recently shown that the tumor microenvironment (TME) plays a vital role in tumor development, including initiation, chronic inflammation, tumor progression, and therapy response. The TME contains tumor-adjacent stromal cells and surrounding immune cells. Among the stromal cells, macrophages have an important influence on tumor progression and affect tumor properties known to be important targets of cancer therapy [6]. 

CD163 is commonly regarded as the receptor of hemoglobin transmembrane scavenger, a specific protein in macrophages, and its expression and gene polymorphism are representatively changed in inflammation [7] and other diseases [8,9]. Therefore, CD163 has great potential to be used as a biomarker to evaluate early inflammatory response, tumor recurrence, and patient survival, among others [10]. One type of macrophage, known as M2 macrophage, is considered to be the main type of tumor-associated macrophage (TAM) and is involved in tumor processes, including development and progression [11,12]. Certain chemicals and cytokines, such as glucocorticoids, IL-6, and IL-10 [13], can stimulate the expression of CD163 in macrophages, whereas other molecules, such as CXCL4 [14], IL-4, lipopolysaccharide (LPS) and TNF-α [12], can suppress its expression. Previous studies have shown that the abnormal expression of CD163 affects tumor development in cancers. A study on lung adenocarcinoma and squamous cell carcinoma showed that a low expression of CD163 was associated with a higher rate of survival [15]. Higher expression of CD163 was observed in metastatic breast cancer compared to primary breast cancer [16], indicating that it may change during tumor progression. Furthermore, its expression has been lower in certain tumor cells compared to its expression in monocytes and macrophages [17,18]. 

Programmed cell death ligand 1 (PD-L1) has been regarded as an efficient biomarker to predict the response to immunotherapy. It can be expressed on the surface of tumor cells and inflammatory cells, and after binding with T cells, activated T cells can be suppressed and inactivated [19]. Anti-PD-L1 immune checkpoint inhibitors are widely used for last-stage malignant tumors, especially metastatic non-small cell lung carcinoma, clear cell renal cell carcinoma, breast cancer, and melanoma [20]. Similar to PD-L1, PDCD-1 has also been used as a biomarker of immunotherapy response [21]. Since the most common assay for these two biomarkers is immunohistochemistry (IHC) [18], we chose IHC to detect the changes in CRC histopathological slices in our study.

The Cancer Genome Atlas (TCGA) and Gene Expression Omnibus (GEO) are two of the major global clinical gene databases. They contain large amounts of information on different types of cancer, such as demographic information, high-throughput gene expression, and DNA polymorphism. By data mining, we can analyze the relationships among the clinical treatments, the regulation of gene expression, the mutation of genes, and the occurrence and development of a specific cancer [19].

To examine the role of CD163 in CRC development, patient survival as well as its relationships with other biomarkers in the present study, we examined CD163 expression in normal and cancer tissues from Swedish CRC patients and further analyzed the relationship between CD163 expression and patient outcomes, including survival. Using different methods of bioinformatics, including gene expression difference analysis, survival analysis, gene set enrichment analysis, and tumor-infiltrating immune cell analysis, and combined with histopathological observation, we discovered the diversity between tumor and normal tissues, the relationship between the expression and mutation of CD163, and its effects on the prognosis of CRC patients. CD163 can therefore be considered a novel biomarker for prognosis, disease progression monitoring, and treatment response.

## 2. Materials and Methods 

### 2.1. Cancer Patients and Immunohistochemistry

This study included 1247 primary CRC patients from several hospitals in Sweden between 1972 and 2014. The clinicopathological data included sex, age, tumor location, TNM stage, differentiation, growth pattern, treatment, survival, and other detailed information. Normal and cancer tissues were used for the experiments to examine the polymorphism and expression of numerous proteins.

IHC for CD163 was performed using 4 μm tissue sections, including normal mucosa, primary tumor, and metastasis in the lymph nodes from paraffin-embedded specimens. The specimens were deparaffinized in Aqua de Par 10 Ancillary reagent (ADP1002 M, BioCare Medical, Pacheco, CA, USA) for 20 min after being kept at 65 °C for 2 h. All sections were then placed into a pressure cooker Decloaking Chamber NexGen configured in a temperature cycle to achieve a maximum of 110 °C for 5 min to perform heat-induced epitope retrieval using Borg Decloaker RTU antigen retrieval solution (BD1000). The slides were then rinsed in tap water before being washed in tris-buffered saline (TBS). Primary antibody against human CD163 (10D6), abcam, Waltham, MA, USA) was fully automated on Leica Bond3 at 1:100 dilution for 15 min; post-primary for 8 min, polymer for 8 min, peroxide for 5 min, DAB chromogen for 10 min, and counterstain HTX for 5 min. The intensity of the brown color was scored by two pathologists separately using -, +, ++, and +++ scores, corresponding to negative, slight, moderate, and strong expressions. 

### 2.2. Data Source

In the GDC database of TCGA (download date: 24 May 2022), we selected adenoma and adenocarcinoma samples in the colon and rectum (TCGA-COAD and TCGA-READ) to obtain gene expressions (FPKM), simple nucleotide variations (SNPs) and clinical data. In this study, we downloaded 472 cases of gene expression data (COAD: 382, READ: 90), and the number of files was 530, including 488 cancer samples and 42 samples adjacent to cancer.

Stemming from the GEO database, as shown in Table 1, we collected a total of 964 cases from 5 GSE datasets. 

### 2.3. Difference Analysis and Intersectional Genes

We performed difference analysis on the FPKM data of colorectal cancer in TCGA. We searched the differentially expressed genes (DEGs) between the tumor tissues and normal tissues with the Wilcoxon rank-sum test using R [27]. First, we read the mRNA expression data using R, took the average expression values for the repeated gene names, and obtained the expressions of 19,938 genes. Then the data were screened to remove small fluctuation genes in all samples; there were only 13164 genes with a row mean > 0.5. The false discovery rate (FDR) was used to correct the *p*-value and control the false positive rate, and the constraint of this study was |logFC| > 1 and FDR < 0.01. Finally, we made a volcano plot with all differentially expressed genes to give an overview of the difference distribution and intersected the 3177 DEGs with 55 genes from the Swedish clinical database to obtain the intersectional genes for subsequent research. Afterwards, we continued the difference analysis on the expression of the selected genes with gene expression data from 5 GEO databases (GSE20842, GSE44076, GSE83889, GSE87211, and GSE90627) to validate the results.

### 2.4. Survival Analysis

Cox regression is a semi-parametric analysis method that assumes that each risk factor’s role does not change with time [28,29,30]. In this study, the RNA expression data and clinical data (438 cases with survival time ≥ 30 days in TCGA and 562 cases from GSE39582) of intersectional genes were combined and transformed into z values using R. We used the survival package to analyze the combined dataset by univariate Cox analysis and drew a forest map to predict the risk factors using the forestplot package in R. 

With the CD163 expression data and clinical features from Swedish hospitals, we analyzed the relationship between high or low CD163 expression and the 5-year survival rate by performing Kaplan–Meier survival analysis. According to the principle of the Kaplan–Meier method [31], samples were grouped according to the binary situation of the target gene expression (up-regulated or not). We compared the number of surviving patients at different time points to determine whether there were differences in the survival of patients. Then, the log-rank test [32] was performed to ensure a statistical difference between groups by calculating the *p*-value. The survival curve of CD163 was drawn by using the survminer package in R.

### 2.5. Enrichment Analysis and Co-Expression Gene Identification 

GSEA is an analysis method used to obtain an enrichment score by calculating the correlation between phenotype and each molecular mechanism [33]. A total of 488 CRC samples were divided into high and low groups by the median CD163 expression, and the log2FC for each group was calculated. After ranking by Log2 FC, we performed GSEA enrichment with the KEGG database using the fgsea package [34] in R (*p*-value < 0.01). The ten highest related biological processes were drawn by the enrichplot package in R. After that, we used the GSVA algorithm to evaluate the score of hallmark gene sets. Wilcox’s rank test was utilized to assess the significance.

Then, according to the Pearson correlation coefficient for CD163 and 13163 other genes in TCGA, 228 genes were selected as the co-expression with CD163 (|Pearson coefficient| > 0.6). A protein–protein interaction (PPI) network was constructed for these co-expression genes related to CD163 using the STRING database [35], with a combined score > 0.7. The PPI network was then put into Cytoscape software [36]. Several clusters were recognized by the default algorithms of the MCODE Cytoscape plugin [37], and the appearance was modified with different colors. Focusing on clusters involving CD163, the clusterProfiler package in R was used to explore the function of this cluster. For further investigation of the CD163 cluster, we made a waterfall plot to visualize the mutation with the maftools package [38] in R and analyzed the relationship between CD163 expression and the mutation of this cluster.

### 2.6. Tumor-Infiltrating Immune Cell Analysis 

To further explore the correlation of CD163 expression with the immune microenvironment, we applied GSVA analysis and CIBERSORT computational methods [39] to evaluate the progression level of tumor-infiltrating immune cells (TICs) in COAD and READ cases from the TCGA database. For GSVA, we analyzed 530 cases and drew a heatmap to reveal the modification of 28 TICs in different CD163 expression groups by using the GSVA package [40] in R. According to the CIBERSORT algorithm, we calculated the infiltrating fraction of the 22 kinds of immune-cell subsets using the Kruskal–Wallis H test based on 530 TCGA cases. This inferred a possible correlation between the expression of CD163 and the progression of lymphocyte infiltration in CRC tissues. The ESTIMATE algorithm [41] was used to further estimate the situation of immune cell infiltration and indicated the level of tumor purity. A higher immune score refers to low tumor purity and the high infiltration of lymphocytes. Additionally, we investigated the relationship between the expressions of representative biomarkers for immune checkpoint inhibitors, PD-L1 and PDCD-1, and the expression of CD163 in CRC patients. The PD-L1 and PDCD-1 expression data were obtained from the TCGA database and divided into normal, high, and low expression groups for comparison. 

## 3. Results 

### 3.1. Screening of Potential Biomarkers

The workflow of our research process is described in Figure 1. We used the TCGA and GEO databases and clinical experimental data to screen potential biomarkers through differential analysis of gene expression. Univariate Cox analysis and the K-M curve were used for the survival analysis of CRC patients. For further exploration of the correlation between target genes and the immune microenvironment, we performed tumor-infiltrating immune cell analysis using the CIBERSORT algorithm, GSVA analysis, and ESTIMATE algorithm. Finally, protein–protein interaction networks and pathway enrichment analysis were used to investigate the relationship between CD163 and colorectal cancer.

### 3.2. Expression of CD163 in CRC Patients and Survival 

As shown in Figure 2, CD163 was strongly expressed in the macrophages of the intercellular matrix of both normal and CRC tissues. CD163 was negatively to slightly expressed in the normal epithelial cells (a, b, c, and d). However, CD163 was negatively (e and f), moderately (g and h), and strongly (i and j) expressed in the cytoplasm of the cancer cells.

### 3.3. Difference Analysis Result 

According to the screening conditions in this study (|logFC| > 1 and FDR < 0.01), we obtained 3177 DEGs, of which 1942 were up-regulated genes and 1235 were down-regulated genes in CRC. Using a volcano map of all DEGs (Figure 3a) and crossing the differential analysis results with genes from clinical experiments, we found four intersectional genes (PCNA, LOX, BCL2, and CD163); see Figure 3b. We then mapped a box plot of all the intersectional genes (Figure 3c–h) based on the five datasets. We found that PCNA was highly expressed in CRC tissue in all groups except GSE20842, where no statistical significance was found. LOX was up-regulated in all groups except for GSE20842 and GSE90627, where no statistical difference was found. In contrast, BCL2 and CD163 had low expressions in the tumor groups in all databases.

### 3.4. Survival Analysis

This study used R to perform the univariate Cox analysis based on gene expression data. We calculated and presented each gene’s hazard ratio and 95% confidence interval (Figure 4a,b). The results show that, for the TCGA database, only CD163 (*p*-value = 0.02) had statistical significance, and for GSE39582, there were three genes: CD163, LOX, and BCL2, with statistical meaning. According to Figure 4, BCL2 might be considered a good prognostic factor for colorectal cancer patients, while LOX and CD163 might be worse prognostic factors.

The CD163 expression data were divided into four groups: tumor tissue with high or low cellular expression, normal tissue with different cellular expression, tumor tissue with high/low expression in the stroma, and normal tissue with different stroma expressions. The data were generally either derived from inside cells or from the adjacent stroma. The results are shown in Figure 4c–f. After analyzing the clinical data, we found that for the cellular group patients, those with up-regulated CD163 had a much lower cumulative survival probability within 5 years compared to those with down-regulated CD163 after censored selection. The log-rank test also confirmed that this trend had a significant statistical difference. Similarity was also observed in the stroma group, and the tendencies of the curves show the same pattern with the log-rank *p* < 0. 05. For the normal groups, there was no statistically significant difference (Log-rank *p* > 0.05) between the low and high expression groups, but the trend had a pattern similar to that in the tissue groups, which indicates that a low expression was related to better prognosis.

### 3.5. Tumor-Immune Microenvironment Analysis Results 

As per all RNA-seq samples in the TCGA database, infiltration progression was estimated and visualized by heatmap using ssGSEA evaluation (Figure 5a), reflecting the expression levels of 22 TICs in each sample. In the bar chart (Figure 5b), the proportion of TICs in three CD163 expression groups corresponded to the infiltration level. The results show that except for memory B cells, CD4 memory resting T cells, regulatory T cells, gamma delta T cells, activated dendritic cells, and eosinophils, other subtypes of immune cells’ proportions were remarkably diverse among the three groups in colorectal cancer tissues. To further investigate the potential association between CD163 expression and TME, we used the ESTIMATE algorithm and retrieved the scores from various dimensions (Figure 5c). Both of the CD163 dysregulated expression groups reflected low immune scores compared to the normal expression group, which suggests low immune cell infiltration. For the stromal score, there was no statistical difference between high expression and normal expression, while the CD163 down-regulation led to a low stromal score. The estimate score is equal to the immune score plus the stromal score and showed the same pattern as the other two components. The estimate score indicates that CD163 expression affected TME significantly. All results are in accordance with the conclusion of the tumor purity score. For the PD-L1 and PDCD-1 expressions, high CD163 expression could induce the expressions of both biomarkers, and low CD163 expression suppressed the expressions of PD-L1 and PDCD-1. Based on these findings on tumor purity and immune escape, we propose that different CD163 expression groups respond differently to immunotherapy in CRC.

### 3.6. Enrichment Analysis of CD163

As shown in Figure 6, there were some highly related molecular alterations and signaling hallmark pathways with CD163 dysregulation. The top 10 biological functions based on the GSEA results show that CD163 played a role in cytokine interactions, extracellular matrix (ECM) receptor interaction, focal adhesion, hematopoietic cell lineage, intestinal immune network, leishmania infection, systemic lupus erythematosus, toll-like receptor signaling pathways, cell adhesion molecules cams, and chemokine signaling pathways (Figure 6a). To investigate the impact on hallmark pathways, we performed GSVA enrichment analysis. CD163 affected 14 hallmark pathways, mainly revolving around inflammatory response and neoplasia. Figure 6c,d shows the protein–protein interaction network. After GO enrichment analysis, 14 genes (including CD163), presented as the whole cluster linked to immune receptor activity in the network, are highlighted, and genes interacting with CD163 are also shown. In Figure 6e, the frequency and type of mutation in the CD163 cluster are illustrated; 19.91% of 462 barcode mutations are located in this cluster. Therefore, we propose that CD163 is more commonly mutated in CRC patients.

## 4. Discussion

In this study, we analyzed the expressions of CD163 and other genes in CRC and evaluated their relationships to clinical outcomes in CRC patients. After analyzing data from public databases using gene expressional difference analysis, we found that CD163 expression in CRC was lower compared to normal tissue. Up-regulated CD163 expression seemed to be a negative prognostic factor for CRC based on the Cox analysis, which was also confirmed by survival analysis based on clinical CD163 expression data. Enrichment analysis and protein–protein interactions networks showed that CD163 was related to several molecular functions and hallmark pathways. Thus, based on these findings, we propose that CD163 could be regarded as a biomarker for tumor progression and clinical outcomes in CRC, where a low expression may be seen in the early stages of development, and high expression may suggest invasion, metastasis, and a low survival rate. In accordance with this study, our previous studies on CD163 have also shown positive expression in approximately 20% of CRC patients and its relationship to early local recurrence, short survival time, decreased apoptotic activity [9], and advanced tumor stage [42].

The TME has a vital influence on tumor progression and prognosis. TAM is one important component of the TME and plays a crucial role in the infiltration, invasion, and metastasis of tumors. Therefore, specific biomarkers on TAM, such as CD163, can be highly clinically relevant to evaluate the response to the treatment of cancer [43]. According to Meisel et al.’s [44] study on CD163-related spatial matrix, the distance between CD163^+^ TAM cells and the tumor cells and the density of CD163^+^ TAM cells could be considered independent predictive indicators for the prognosis of cancer. A higher density of CD163^+^ TAM cells and a shorter distance could be regarded as a clinical and pathological risk factor for breast cancer. Another study by Krijgsman et al. [45] showed the expression of CD163 stemming from three different components in CRC patients, including plasma, monocytes, and macrophages. Their findings showed that high levels of CD163 expression in serum were associated with worse survival. As per the previous study on macrophage infiltration in CRC, CD163 expression was higher in tumor tissues with high macrophage infiltration and associated with worse prognosis [42]. Following these clues, using the human expression data from public databases, we further analyzed the role of CD163 in TME and investigated the relationship between CD163 expression and 22 types of TILs. Furthermore, the estimate scores and tumor purity indicate that there was a significant difference between the low and high CD163 expression groups, where a low expression of CD163 was related to higher tumor purity. After investigation of the expressions of PD-L1 and PDCD-1, which are biomarkers for immunotherapy response, against CD163 expression, we found that the dysregulation of CD163 expression was associated with statistically different PD-L1 and PDCD-1 expressions. Hence, we propose that patients with different CD163 expressions have different levels of sensitivity towards anti-PD-1L or anti-PDCD-1 immune checkpoint inhibitor therapy. Some previous studies have already discussed the possibility of using CD163 as a diagnostic and therapeutic biomarker in some malignant diseases [42,43,44,45,46,47,48]. A high expression of CD163 is seen as an indirect marker of monocyte and macrophage activation [49]. Therefore, most studies focusing on the dysregulation of CD163 expression in inflammatory response revolve around the immunological properties of CD163 [50]. Meanwhile, CD163^+^ TAMs are also known to be associated with epithelial–mesenchymal transition (EMT), the mesenchymal circulating tumor cell ratio, and a poor prognosis of CRC [51], in accordance with this present study. However, one study showed that an increased number of CD163-positive glioma-associated microglia/macrophages in the tumor core was related to better survival [52]. However, the study used transcriptome analysis, while we used immunohistochemistry to examine the expressions. Therefore, the expressions were analyzed at the mRNA level and the protein level, respectively. This methodological difference and the different phenotypes of CD163^+^ TAM [44] might have led to the difference in results. It might also be suggested that the relationship between CD163 and the survival rate is different in certain specific cancer types, such as gliomas. Our study provides another piece of convincing evidence to support the effects of CD163 in CRC as well as other types of cancer. Further experiments are needed to further confirm the specific regulation at different stages of cancer and the roles and specific functions of CD163 in different pathways.

Compared to previous findings, our study is the first to combine data from several huge public databases with experimental data to study the possible function and effects of CD163, focusing on TME and TILs in CRC patients. Previous studies have performed their analyses on CD163 related to the number, distribution, and phenotype comparing tumor and normal tissue/patients. We have deepened the understanding of the influence of CD163 on CRC prognosis and also the possible effects on 22 types of lymphocytes, tumor purity, immune check-points such as PD-L1, and the potential involvement of CD163 in different CRC-relevant cancer hallmark pathways and biological processes. Limitations in this study mainly revolve around the raw data from the hospitals. Certain gene expression data were not able to be retrieved, which may influence the accuracy, precision, and power of the statistical analyses. Reasons for the lack of certain gene data might be due to poor follow-up, old experimental methods, or outdated information systems. In our study, although the overall number of cancer cases was enough to obtain good-quality results, more data and cases about CD163 are still needed because of unequal distribution after grouping based on age and sex, which may explain the lack of differences in the survival analysis. This study showcases the function and multiple relationships of CD163 in CRC to further define the value of CD163 as a potential biomarker in the prognostic prediction and assessment of CRC. We analyzed CD163 from three dimensions: genomics, histopathology, and epidemiology, which increases its relevance to real disease. Studies on the downstream and upstream regulators of CD163 and comprehensive mechanical mapping of CD163 are needed.

## 5. Conclusions

In this study, we examined CD163 expression in normal and cancer tissues using multiple cohorts of patients and investigated how the dysregulation of CD163 was associated with tumor TME and TICs, as well as patient prognosis at the level of transcriptome and proteome. Based on these findings, we propose that CD163 could be used as a biomarker for evaluating CRC development and determining the prognosis of patients. The results may even provide data needed for further research on developing valuable CRC surveillance programs and designing efficient therapies for patients.

## Figures and Tables

**Figure 1 cancers-14-06166-f001:**
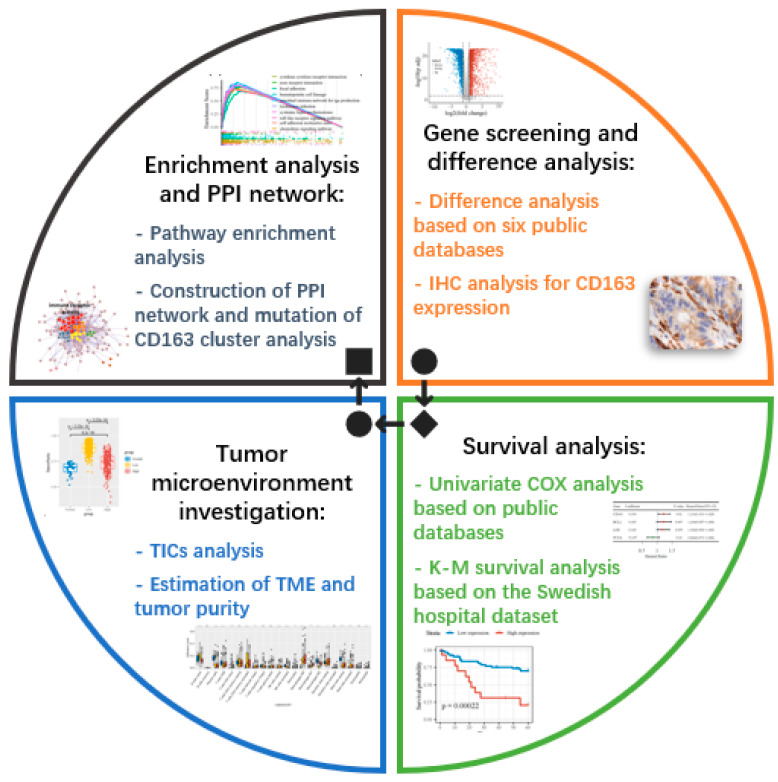
Research process of this present study. The structure and workflow of this investigation were divided into four parts: gene screening and difference analysis, survival analysis, TME investigation and enrichment analysis, and construction of PPI network. Data mining was based on six public databases to find biomarkers with significantly different expressions, which were confirmed by tissue micro-array samples from CRC patients. As per survival data, two different methods (Cox and K-M survival analysis) were performed to determine the relationship between our factor and patient prognosis. Following the first two steps, the analysis of TME and enrichment analysis have been applied in this study to ensure the role of CD163 in TME and its possible pathways.

**Figure 2 cancers-14-06166-f002:**
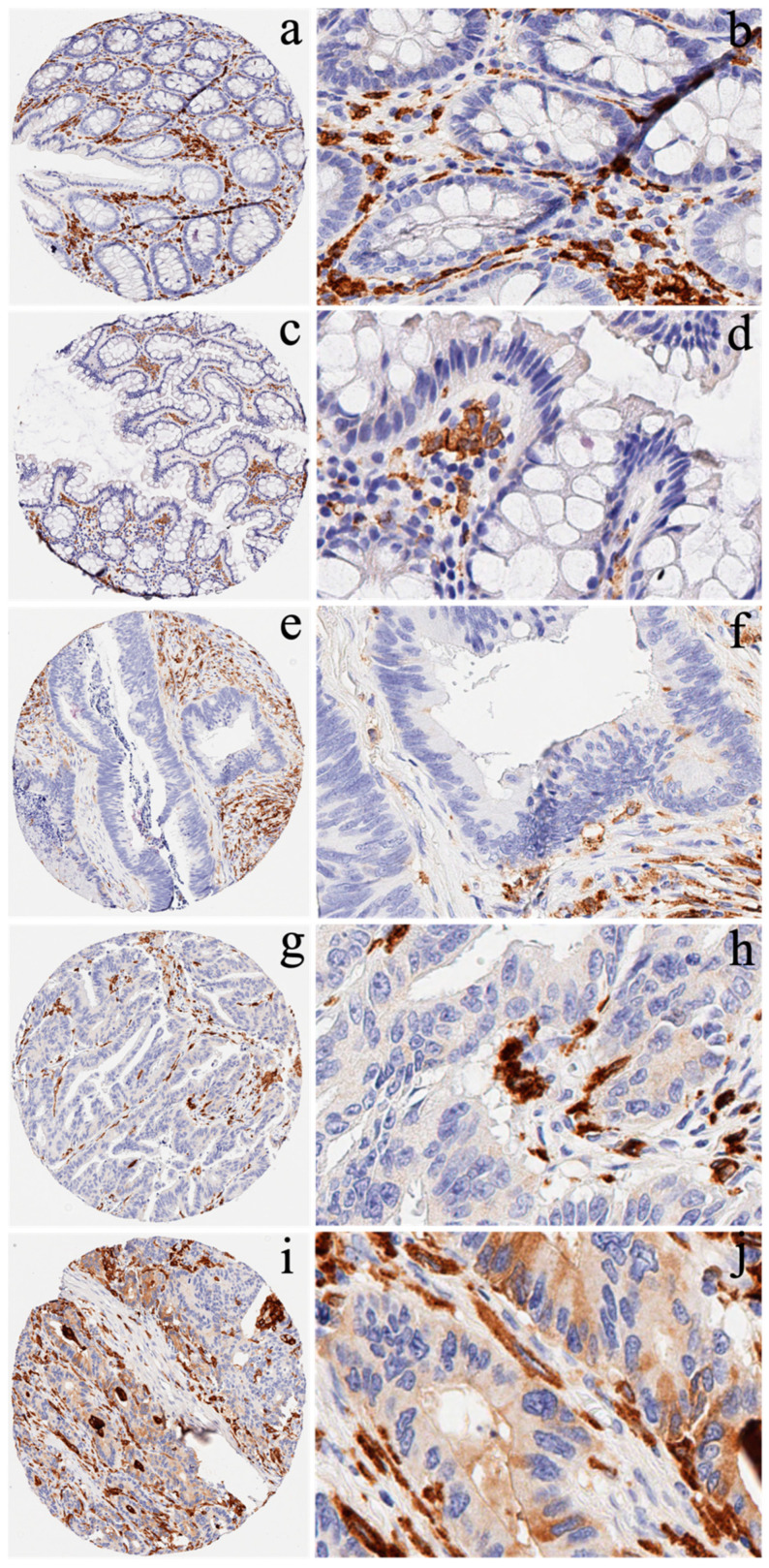
Expression of CD163 in normal colorectal mucosa and cancer tissue using IHC. CD163 was strongly expressed in the macrophages of normal and cancer intracellular matrices. CD163 was negatively (**a**): low magnification (×10) and (**b**): high magnification (x40) and slightly expressed (**c**): low magnification (×10) and (**d**): high magnification (×40) in the colorectal epithelial cells. However, CD163 was negatively (**e**): low magnification (×10) and (**f**): high magnification (×40), moderately (**g**): low magnification (×10) and (**h**): high magnification (×40), and strongly (**i**): low magnification (×10) and (**j**): high magnification (×40) expressed in the cytoplasm of the cancer cells.

**Figure 3 cancers-14-06166-f003:**
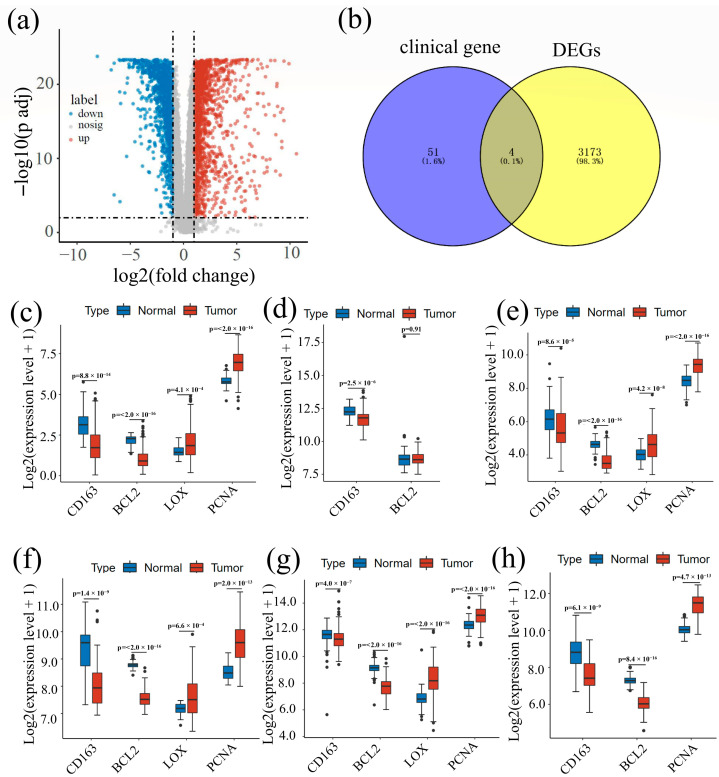
(**a**) Volcano plot of all differentially expressed genes in TCGA; red means up-regulation, blue means down-regulation, and gray means interferential genes. (**b**) Venn diagram for the intersection between genes from TCGA database after difference analysis (yellow part) and genes recorded in hospital database (blue part). (**c**–**h**) Box plots of intersectional gene expressions. Based on the significant differences among the expressions of four genes, the gene expression was calculated to log2 (gene expression + 1) to show the difference between each gene. (**c**) Expression data from TCGA; (**d**) expression data from GSE20842; (**e**) expression data from GSE44076; (**f**) expression data from GSE83889; (**g**) expression data from GSE87211; (**h**) expression data from GSE90627.

**Figure 4 cancers-14-06166-f004:**
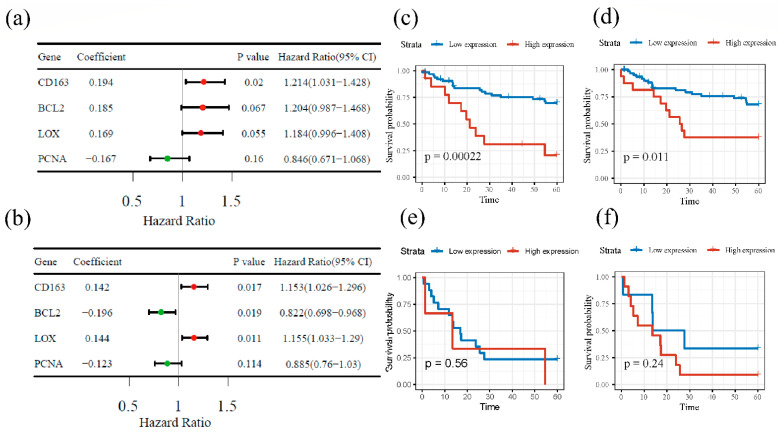
Forest map of four intersectional gene hazard ratios and confidence intervals (CI). Green block means positive prognostic factor, and red means negative prognostic factor. (**a**) Data from TCGA and (**b**) data from GSE39582. K-M survival curves show CD163 expression in relation to survival probability of colorectal cancer patients from several Swedish hospitals. (**c**) K-M survival analysis of tumor-tissue cellular CD163 differential expression; (**d**) K-M survival analysis of tumor-tissue stromal CD163 expression; (**e**) K-M survival analysis of normal-tissue cellular CD163 expression; (**f**) K-M survival analysis of normal-tissue stromal CD163 expression.

**Figure 5 cancers-14-06166-f005:**
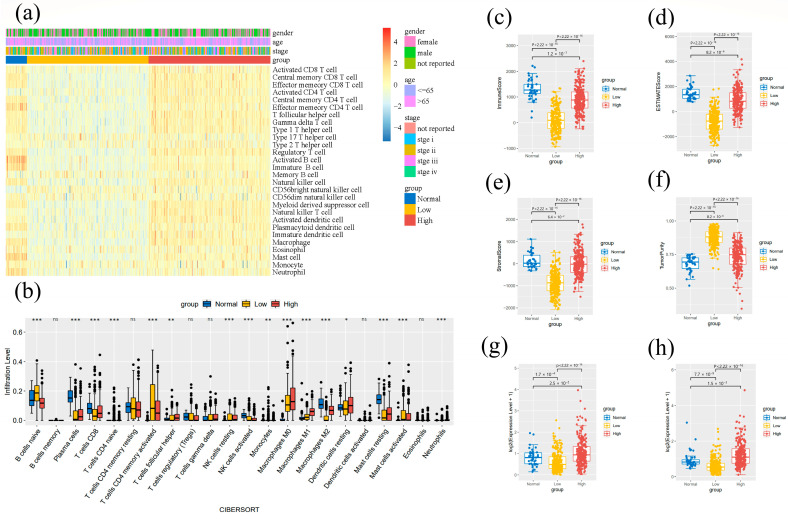
Evaluation of CD163 expression related to TME based on colorectal cancer cases in the TCGA database. (**a**) Heatmap of 28 TICs divided by differences in CD163 expression. Red means high expression, blue means low expression. Gender, age, and tumor stage indicated by legend. (**b**) Box plot showing the proportions of 22 types of lymphocytes associated with CD163 expression (* *p*-value < 0.05, ** *p*-value < 0.01, *** *p*-value < 0.001, ns represents no significance: *p*-value > 0.05). (**c**–**f**) The immune score, estimate score, stromal score, and tumor purity of different CD163 expression groups. (**g**) The relationship between the expression of PD-L1 and CD163 expression across normal, low, and high expression groups. (**h**) The relationship between PDCD-1 expression and CD163 expression among normal, low, and high expression groups.

**Figure 6 cancers-14-06166-f006:**
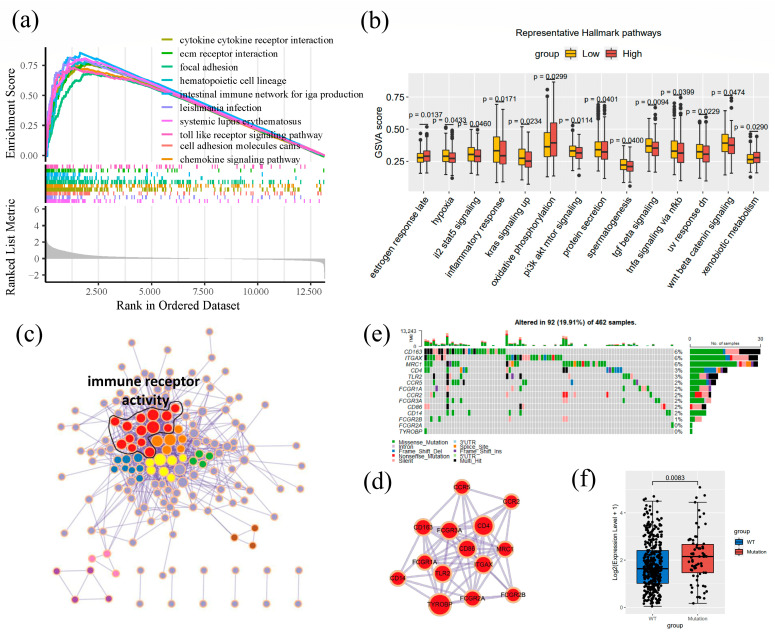
Analysis of potential biological processes associated with CD163. (**a**) Top 10 pathways based on GSEA enrichment results of CD163. (**b**) Difference between CD163 high and low expression groups in several hallmark pathways. (**c**) Gene enrichment results in protein–protein interaction (PPI) network of colorectal cancer. (**d**) Modules involving CD163 in PPT network. (**e**) Mutations of all genes from CD163 module in PPI network. (**f**) The relationship between CD163 expression and mutation of CD163 module in PPI network.

**Table 1 cancers-14-06166-t001:** Information on the included GEO datasets.

Accessions	Platforms	Samples (Tumor vs. Non-Tumor Tissues)	References
GSE20842	Agilent-014850 Whole Human Genome Microarray 4 × 44 K G4112F (Feature Number version)	65 vs. 65	PMID: 20725992 [22]
GSE44076	Affymetrix Human Genome U219 Array	98 vs. 148	PMID: 25215506 [23]
GSE83889	Illumina HumanHT-12 V4.0 expression beadchip	101 vs. 35	PMID: 28455965 [24]
GSE87211	Agilent-026652 Whole Human Genome Microarray 4 × 44 K v2	203 vs. 160	PMID: 29119627 [25]
GSE90627	Agilent-039494 SurePrint G3 Human GE v2 8 × 60 K Microarray	32 vs. 96	PMID: 28977850 [26]

## Data Availability

The clinical datasets of patients with primary CRC in some hospitals in Linköping that were analyzed or generated during the study are available from the corresponding author upon reasonable request. The RNA-seq data and SNP data used in this study can be obtained in the GDC databases of TCGA and GEO.

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
