# Peer review of "CD163 as a Potential Biomarker in Colorectal Cancer for Tumor Microenvironment and Cancer Prognosis: A Swedish Study from Tissue Microarrays to Big Data Analyses"

_cancers, 2022, doi:10.3390/cancers14246166_

Round 1
Reviewer 1 Report
CD163 as a potential biomarker in colorectal cancer for tumor microenvironment and cancer prognosis: a study from tissue microarrays to big data analyses
In this study, the authors presented data extracted from a large number of colon cancer patients' tissues from Sweden. By taking advantage of huge databases and employing some software, they were able to come out with promising results. Here are some of my comments on the study.
Title:
Since the study is limited to patients from Sweden, it is better to specify this in the study title. Because ethnic diversity has become a prominent challenge in the field of diagnosing genetic diseases, specifically cancerous.
Introduction:
Authors should begin the introduction to the manuscript with the reader's most important question. Here they started talking about cancer statistics and the cancer epidemic and the importance of studying it. I think the priority here is to get straight to the point.
Methods:
The method did not contain functional studies of this gene either in vivo or in vitro. This shortcoming reduces the power of the study.
Results:
The specificity of any diagnostic tools is extremely important. Is the specificity of this gene sufficient in these data? What about healthy population.
Performing some molecular experiments is highly recommended such as PCR and Western blot. This will give the results more strength.
Discussion
I think the authors do not discuss the results in sufficient depth. There are published studies on gene CD163 and its relationship to colon cancer. It is not discussed here or addressed even briefly. Here are some examples:
Maisel, Brenton A., et al. "Spatial Metrics of Interaction between CD163-Positive Macrophages and Cancer Cells and Progression-Free Survival in Chemo-Treated Breast Cancer." Cancers 14.2 (2022): 308.
Krijgsman, Daniëlle, et al. "CD163 as a biomarker in colorectal cancer: The expression on circulating monocytes and tumor-associated macrophages, and the soluble form in the blood." International journal of molecular sciences 21.16 (2020): 5925.
Author Response
- title:
Response: Following the reviewer’s suggestion, the title has been changed to “CD163 as a potential biomarker in colorectal cancer for tumor microenvironment and cancer prognosis: a Swedish study from tissue microarrays to big data analyses.”
- introduction:
Response: This is very valuble suggestion. We have modified the introduction and now addressed the reader’s most important questions in the beginning . Reference 3 has been updated to use the newest data from 2020 instead of 2017 and other references have been adjusted as well in the revised MS.
- methods:
Response: In this project, we mainly focuced on the study of CD163 protein expression in the tissue microarray samples from the colorectal cancer patients and the relationship with clinical outcomes including survival. We have further enhanced the power of CD163 as a potential biomarker by utilizing the following bioinformatic approaches, and all available data is included in the revised MS.
1). The expression levels of 1533 colorectal cancer samples from 6 different datasets;
2). The relationship between the expression of CD163 and prognosis at the level of transcriptome and proteome was analyzed;
3). We have also performed immune infiltration assessment using multiple algorithms including ESTIMATE, CIBERSORT and GSVA to comprehensively evaluate the samples;
4). We have deeply explored the possible mechanisms by which CD163 affects immune infiltration at the level of multiple omics.
- results:
Response: Thank the reviwer’s comments. In this study, we focuced on CD163 protein expression in the tissue microarray samples from colorectal cancer patients and the relationship to progression of the cancer. Regarding the issue of research specificity, our study used multiple samples from the TCGA and GEO databases to make our data more diversified. And then, from various angles including transcriptomics, proteomics, and other levels of analysis analyze the role of CD163 in patients with colorectal cancer. The above content reflects the specificity of this study. We also used cancer and adjacent tissues for differential analysis and applied patient data from Swedish hospitals. If the later conditions are mature, we will verify it in the healthy population. In this part of the experiment, this study used the immunohistochemistry experiment to detect the protein polymorphism and expression in normal and cancer tissues of colorectal cancer patients. We will keep reviwer’s valuable opinion in our mind to collect health individuals in the future, and will carry out molecular experiments to validate the specificity of CD163 gene as an earlier dgnostic marker.
- discussion:
Response: Thank the reviwer’s suggestions. We have carefully checked these valuable studies and added the associations of the tumor cells-CD163+TAM distance with the cancer prognosis, and the dysregulation of CD163 expression in different components like serum, tumor-associated macrophage and monocytes to in the discussion section of the MS. We have also citated these two important references in the revised MS.
Reviewer 2 Report
In this manuscript, Ma et al. present a study on CD163 as a potential biomarker in colorectal cancer for tumor microenvironment and cancer prognosis: a study from tissue microarrays to big data analyses. The authors combined analysis of tissue microarray (TMA) samples from colorectal cancer (CRC) patients with bioinformatical analysis of some public databases and the clinical dataset and identified the different expression of CD163 in various tissues, the presence of the receptor in TME, the interaction with other biological processes and a positive correlation between CD163 dysfunction and worse prognosis. Their results indicated that CD163 has the potential to be a predictive biomarker in the investigation of CRC. This manuscript is interesting, and the experiments are well thought out and expertly executed, I think it can be accepted for publication.
Author Response
Thank you very much for your perfect understanding of our study, we appreciate your wonderful comments and positive suggestion of its acceptance.
Reviewer 3 Report
This work deals with CD163 as a potential biomarker in colorectal cancer for tumor by Ma et al. The work seems average and need major over hauling before acceptance. The following points will make it attractive article.
1. The addition of the latest data n world cancer scenario.
2. The addition of notorious effect of cancer and the authors may consult ollowing refs.
RSC Adv., 4, 29629 - 29641 (2014).
3. Please improve the quality of the Figures as many figures are not redable.
4. The concussion should be concise and to the points indicating the application of the work.
5. English should be improved throughout the manuscript.
6. Quantitative information should be provided in the abstract.
8. Please provide error graphs in the figure; where are required.
9. Please compare your results with previous studies and mention clearly how your work is important in comparison to already been reported.
Author Response
1. We have revised the manuscript to cite the latest colorectal cancer data (2020) instead of using the older data from 2017.
2. We have modified the introduction according to this reference citing it as reference 4.
3. Following the reviewer’s suggestion, we have provided high resolution versions of Figures 4-6, and even attached our original Figures as a zip.file in the revised MS.
4. Thank you for your suggestion, we have summarized our work more succinctly in the conclusion in the revised MS.
5. The English in the MS has been carefully checked and revised.
6. Thank you for the suggestion. We have added quantitative information such as p-values in the abstract.
8. Fowllinng valuable suggestion, we have modified all the boxplots in this study and added error bars to show the 95% confidence intervals in the Figure 3 (c-h), Figure 5 ( b-h ), and Figure 6 (b and f) in the revised MS, respectively.
9. Thank you for your suggestion. We have added new comparisons to previous studies in our discussion and reflect on what our work adds to advance the knowledge of CD163 as a potential biomarker in the revised MS.
Round 2
Reviewer 3 Report
Accepted